# A Time to Get Vaccinated? The Role of Time Perspective, Consideration of Future Consequences, Conspiracy Beliefs, Religious Faith, Gender, and Race on Intention to Vaccinate for COVID-19 in the United States

**DOI:** 10.3390/ijerph20043625

**Published:** 2023-02-17

**Authors:** Lening A. Olivera-Figueroa, Alexander Unger, Julie Papastamatelou, Philip G. Zimbardo

**Affiliations:** 1Department of Psychology, Golden Gate University, 536 Mission St., San Francisco, CA 94105, USA; 2Family Medicine Residency Program, AltaMed Institute for Health Equity, AltaMed Health Services Corporation, 2040 Camfield Avenue, Los Angeles, CA 90040, USA; 3East-Asia Institute, Ludwigshafen University of Business and Society, Rheinpromenade 12, 67061 Ludwigshafen, Germany; 4Study Program of Business Psychology, University of Applied Management Studies (HdWM), 68163 Mannheim, Germany; 5Department of Psychology, Stanford University, Jordan Hall 450 Jane Stanford Way, Building 420, Stanford, CA 94305, USA

**Keywords:** time perspective, balanced time perspective, COVID-19 vaccine, consideration of future consequences, conspiracy beliefs, religious faith, gender, race, United States, theory of planned behavior, intention

## Abstract

The present study examined the predictability of Time Perspective (TP) tendencies (i.e., Past Positive, Past Negative, Present Hedonistic, Present Fatalistic, and Future), the Balanced Time Perspective (BTP) profile, the Consideration of Future Consequences—Immediate (CFC-I) factor, the Consideration of Future Consequences—Future (CFC-F) factor, conspiracy beliefs about COVID-19 being a hoax, religious faith, gender, and race on COVID-19 vaccination intention as a dependent variable. Participants were recruited in the United States through the online platforms Prolific and Google Forms. The final sample was *n* = 232 (*n* = 99 male, *n* = 129 female, and *n* = 2 other, M*_age_* = 31). Outcome measures included sociodemographic questions, the Zimbardo Time Perspective Inventory—short version, the Consideration of Future Consequences (CFC) ultra-short scale, the COVID-19 Conspiracy Beliefs questionnaire, and the Santa Clara Strength of Religious Faith Questionnaire—brief version. Regression analyses revealed that vaccination intention was reduced by gender identification as woman, identification as multiracial or from mixed origin, Past Positive, Deviation from a BTP profile, belief in COVID-19 as hoax, and religious faith. Conversely, intention to vaccinate against COVID-19 was increased by Past Negative, CFC-I, and CFC-F. These findings could be beneficial for knowledge transfer to behavioral interventions aimed to promote vaccination against COVID-19, health promotion campaigns, and the public health field.

## 1. Introduction

### 1.1. Impact of the COVID-19 Pandemic and Importance of Vaccination for COVID-19

Over 638 million individuals throughout the world have been infected during the severe acute respiratory syndrome coronavirus 2 (SARS-CoV-2) pandemic, also known as COVID-19, resulting in over 6.62 million confirmed deaths [1,2]. This pandemic has severely affected the world’s economy and healthcare systems [3,4]. In terms of prevention, the most promising method of intervention for COVID-19 is a vaccine. Thus, the global crisis caused by this pandemic led to an unprecedented race to develop a vaccine for COVID-19 [5,6,7,8]. Nonetheless, little was known about how accepted this vaccine would be in the community once it was made available.

### 1.2. Vaccine Hesitancy

Vaccine hesitancy refers to hindrance in the cooperation of individuals on vaccinating themselves, regardless of vaccine availability [9]. Unfortunately, meta-analyses have shown that vaccine hesitancy has appeared to be a major barrier to COVID-19 vaccine acceptance across the globe [10,11]. Thus, it is important to identify the causes of this hesitancy which serve as barriers of COVID-19 vaccine uptake in the community [12]. Likewise, it is also imperative to identify the factors serving as facilitators of positive intentions toward the uptake of COVID-19 vaccines [12,13]. One approach to study barriers and facilitators of COVID-19 vaccination intention is through theoretical frameworks utilized to better understand vaccine hesitancy. 

#### Theoretical Frameworks Previously Utilized to Study COVID-19 Vaccine Hesitancy

Several theoretical frameworks have previously been utilized to study the constellations of psychological determinants of COVID-19 vaccine hesitancy, such as Time Perspective Theory, the Health Belief Model, the 5C psychological antecedents of vaccination model, and the Theory of Planned Behavior [14]. For example, a study conducted across elderly individuals over 65 years old in Italy identified a predictive role of one Time Perspective Theory construct on the intention to get vaccinated in the future against COVID-19 [15]. Furthermore, a study comparing the Health Belief Model, the 5C psychological antecedents of vaccination model, and the Theory of Planned Behavior in Bangladesh identified Theory of Planned Behavior as the model having the highest predictive power of COVID-19 vaccine hesitancy [14]. As the referenced studies suggest, both Time Perspective Theory and Theory of Planned Behavior represent effective models for studying COVID-19 vaccine hesitancy. Having said that, for the present study, Time Perspective Theory represents the main overarching theoretical perspective framing the study we conducted, while Theory of Planned Behavior represents the construct used to measure intention to vaccinate against COVID-19 in our study, as an outcome, dependent variable.

### 1.3. Time Perspective

Time Perspective (TP) refers to the psychosocial, cognitive, and emotional process through which individuals focus positively or negatively on the past, present, and future [16,17]. This construct is most often measured through the Zimbardo Time Perspective Inventory (ZTPI), which many consider to be the “gold standard” TP instrument [16]. The ZTPI is comprised of five scales addressing different temporal tendencies. The five ZTPI scales are: Past Negative, Past Positive, Present Fatalistic, Present Hedonistic, and Future. 

#### 1.3.1. Time Perspective Tendencies and the COVID-19 Pandemic

Of the traditional five TP tendencies, Past Negative refers to an aversive focus on past experiences. During the first weeks of the COVID-19 pandemic in spring 2020, Past Negative positively predicted anxiety in Argentina, France, Greece, Italy, Japan, and Turkey, as well as positively predicted depression in all those countries except Japan [18]. Additionally, a study conducted across individuals of Polish nationality revealed that Past Negative was positively correlated with all forms of conspiracy beliefs about COVID-19 [19].

Past Positive refers to an optimistic, nostalgic view of the past, which has previously been shown to predict increased life satisfaction in Spain [20]. During the COVID-19 pandemic’s first phase of confinement, Past Positive negatively predicted anxiety in Turkey, as well as negatively predicted depression in Argentina, France, Greece, Turkey, and Japan [18]. 

Present Fatalistic refers to a pessimistic view of the present, which when taken to an extreme, has been shown to increase vulnerability toward stress-related problems like allostatic load in Canada prior to the COVID-19 pandemic [21]. During the initial phase of the COVID-19 pandemic, Present Fatalistic positively predicted anxiety in France, Greece, Italy, and Turkey, as well as positively predicted depression in France and Italy [18]. Additionally, Present Fatalistic was positively correlated with all forms of conspiracy beliefs about COVID-19 in Poland [19]. 

Present Hedonistic refers to a pleasurable focus towards the present, which in extreme proportions, could render individuals vulnerable to experiencing accidents, injuries, addictions, and difficulties in personal and professional aspects [22]. During spring 2020, Present Hedonistic positively predicted anxiety in Japan, and negatively predicted depression in France [18]. Additionally, Present Hedonistic was positively correlated with all forms of conspiracy beliefs about COVID-19 in Poland [19].

Future refers to a focus on planning for the future, which has previously been shown to predict increased life satisfaction in Poland [20]. During spring 2020, Future positively predicted anxiety in Italy and Turkey, and negatively predicted depression in Japan [18]. Additionally, the study conducted in Italy across elderly individuals also reported negative correlations between Future and the assumption of risky behaviors for contagion with COVID-19 [15].

#### 1.3.2. Time Perspective Tendencies and Vaccination Intention 

A study conducted across 139 elderly individuals over 65 years old in Italy reported a predictive role of Present Fatalistic on the intention to get vaccinated in the future against COVID-19, as well as on global beliefs about the vaccines against COVID-19 [15]. Regarding TP’s relationship with vaccination intention for other conditions, one article reported an increasing effect of Future TP on readiness to take the influenza vaccine [23]. 

#### 1.3.3. Balanced Time Perspective

Although the five TP tendencies are considered to be independent from one another, two distinct TP profiles have been identified: Balanced Time Perspective (BTP) and Negative Time Perspective (NTP). Individuals who score high on Past Positive, moderate on Present Hedonistic, moderately high on Future, and low on the Past Negative and Present Fatalistic dimensions are considered to exhibit the BTP profile [24]. A systematic review assessing the empirical relationships of BTP with psychological variables such as well-being, mental health, personality, cognitive functioning, self-control, interpersonal relations, biological correlates, and demographic variables concluded that BTP appears to be an important mechanism of adaptation, particularly with well-being [25]. On the case of well-being, a recent study conducted in Chile reported that, together, BTP and mindfulness promote optimal functioning through psychological well-being by alleviating or reducing discomfort [26]. Concerning discomfort, research conducted on burnout syndrome across employees of large corporations in Poland showed that high levels of BTP were correlated with feelings of personal achievement, whereas lower levels of BTP were correlated with emotional exhaustion [27]. Furthermore, the importance of BTP in preventing burnout was also confirmed across healthcare professionals in Germany [28]. Thus, research evidence suggests that BTP is very positive for overall health, both mentally and physically.

#### 1.3.4. Negative Time Perspective

Contrary to the BTP profile, individuals who score high on Past Negative and Present Fatalistic yet low on Past Positive, Present Hedonistic, and Future tend to exhibit a profile opposite to BTP, known as the Negative Time Perspective (NTP) profile [29,30,31]. For example, research conducted in Canada previously showed associations between BTP and amplified cortisol secretion, as well as between NTP and blunted cortisol secretion [32]. Likewise, a study conducted across Puerto Rican populations living in Puerto Rico and Connecticut showed that BTP predicted adaptive coping, whereas NTP predicted maladaptive coping [33]. As such, research evidence suggests that characterization of the NTP profile can be very aversive for overall health, both mentally and physically. 

#### 1.3.5. Balanced Time Perspective, Negative Time Perspective, and the COVID-19 Pandemic

Regarding COVID-19, research conducted in Italy identified BTP as a protective factor that reduces negative beliefs about COVID-19′s consequences on one’s future life, potentially helping those individuals to successfully cope with the COVID-19 pandemic [34]. Similarly, another study conducted in Italy during the mandatory lockdown revealed that individuals characterized by NTP reported lower well-being and more distress during confinement [35]. Most recently, a study conducted in Russia during COVID-19 isolation reported that individuals characterized by BTP had better sleep characteristics, and were less likely to exhibit symptoms of depression, whereas individuals characterized by NTP were more likely to exhibit worse sleep characteristics and symptoms of depression [36]. Additionally, NTP was positively correlated with all forms of conspiracy beliefs about COVID-19 in Poland [19]. Overall, the described studies suggest that individuals focused positively on long-term planning for their lives generally coped better with COVID-19 isolation.

#### 1.3.6. Consideration of Future Consequences

Regarding the future, a questionnaire called Consideration of Future Consequences (CFC) was created separately from the ZTPI, in order to measure the consideration individuals give to the potential distant outcomes of their current behaviors, as well as the extent to which they are influenced by these potential outcomes [37,38]. Theoretically, the CFC construct revolves around an individual’s intrapersonal struggle between current behaviors with a set of immediate outcomes and a set of distant outcomes. Individuals scoring low in CFC tend to focus on the immediate outcomes of their behaviors and therefore behave according to these immediate needs. Conversely, individuals scoring high in CFC tend to be more considerate about the future implications of their behaviors and are expected to act to be more proactive in attaining goals. 

#### 1.3.7. Consideration of Future Consequences and Vaccination

Regarding vaccination, studies conducted in the United States where CFC was treated as a unidimensional construct failed to detect direct effects of CFC on the vaccine uptakes for the human papillomavirus (HPV) and H1N1, yet detected that CFC had a significant indirect effect on HPV and H1N1 vaccine uptakes through perceived vaccine efficacy [39,40]. Pertaining specifically to COVID-19, a recent study also conceptualizing CFC as a unidimensional construct reported positive association between CFC and the willingness to get vaccinated for COVID-19 in France [41]. In contrast to the unidimensional approach to the CFC taken by the aforementioned studies, a study conducted in the United States addressed the predictability of CFC on intentions to vaccinate against COVID-19 and seasonal flu using a two-factor model of CFC [42]. The findings of that study showed that a future CFC factor (labeled CFC-F) predicted vaccination intentions against COVID-19 and seasonal flu, whereas an immediate CFC factor (labeled CFC-I) did not. Furthermore, the same study also reported that both CFC-F and CFC-I had positive indirect effects on vaccination intentions of COVID-19 and seasonal flu through increasing perceived severity of the corresponding disease, but not affective risk perceptions or perceived susceptibility. Since both affective risk perceptions or perceived susceptibility represent beliefs about one’s vulnerability to a disease, that finding was interpreted as indicative that only beliefs about the severity of COVID-19 and flu, but not beliefs about one’s vulnerability to COVID-19 and flu, is what accounts for the influence of CFC-F and CFC-I on vaccination intentions. 

### 1.4. Conspiracy Beliefs about Vaccination for COVID-19

Due to the relevance that beliefs have for intentions to vaccinate for COVID-19, another topic of interest for the current study was conspiracy beliefs about COVID-19. For instance, a systematic review that summarized multilevel determinants of COVID-19 vaccination intention in the United States [43] concluded that belief in conspiracy theories about COVID-19 was a persistent barrier to COVID-19 vaccine uptake [44,45,46,47]. In terms of global variability in vaccine acceptance among different populations across diverse countries and continents, a mixed-method study that used quantitative and qualitative approaches [48] and several systematic reviews [49,50,51] have concluded that the reasons behind vaccine hesitancy and acceptance were similar across the board, with conspiracy theories related to infertility and misinformation about the COVID-19 vaccine on social media often resulting in vaccine hesitancy. Given the vast amount of evidence indicating that COVID-19-related conspiracy beliefs influence intentions to vaccinate for COVID-19 across the globe, such beliefs should be considered in studies addressing multiple predictive factors on hesitancy towards COVID-19 vaccination. 

### 1.5. Religious Faith and Vaccination for COVID-19

In light of the demonstrated impact that beliefs, such as those related to conspiracy, have on intentions to vaccinate for COVID-19, another topic of interest for the current study was religious faith. For instance, one study conducted among adults in Puerto Rico identified associations between greater religiosity and being uncertain or unwilling to get the COVID-19 vaccine [52]. Similarly, religiosity was negatively associated with intention to vaccinate against COVID-19 in mainland United States [53]. Moreover, three recent studies that encompass 195 regions from around the world reported that regions identified as higher in spirituality and/or religiosity were the regions characterized by lower COVID-19 vaccination rates [54].

### 1.6. Theory of Planned Behavior and Vaccination Intention against Illnesses like COVID-19

The Theory of Planned Behavior centers around three types of beliefs that tend to guide human behavior: behavioral beliefs, normative beliefs, and control beliefs [55]. Theory of Planned Behavior predicts that, when combined, attitudes, subjective norms, and one’s perceived behavioral control related to a behavior, lead to the development of an intention to engage in that behavior or not. Understanding these beliefs and the intentions they evoke, can provide insights on how to impact behavior change [55]. Regarding Theory of Planned Behavior and vaccination, a recent meta-analysis found clear support for the utility of Theory of Planned Behavior in explaining vaccine hesitancy generally [56]. Moreover, Theory of Planned Behavior has been used to understand its role in predicting intentions to receive vaccines for diverse illnesses such as the human papillomavirus (HPV) [57], influenza [58,59], and H1N1 [60,61,62]. On the case of HPV, Theory of Planned Behavior consistently outperformed the Health Belief Model [57]. Concerning influenza, the Theory of Planned Behavior predicted intentions to vaccinate for influenza across older adults [58] and children [59]. Regarding H1N1, the Theory of Planned Behavior has appeared to contribute to intended vaccination uptake in one study [60], predicted 60% of adults’ intention to receive an H1N1 vaccination in another study [61], and explained 70% of the variance in vaccination intention in another study [62]. Regarding COVID-19, a recent systematic review and meta-analysis investigated the role of Theory of Planned Behavior constructs in determining intention to get vaccinated against COVID-19 [63]. The findings of this meta-analysis revealed that the COVID-19 vaccination intention rate was relatively high at 73%. In light of these findings, the authors of this meta-analysis concluded that Theory of Planned Behavior indeed serves a useful framework for studying intention to receive a COVID-19 vaccine.

### 1.7. Research Aim of the Study 

The primary aim of this study was to examine the influence of time perspective, consideration of future consequences, conspiracy beliefs about the COVID-19 pandemic, and religious faith on intention to vaccinate for COVID-19 in the United States. 

### 1.8. Research Question

Do TPs, CFC, COVID-19 conspiracy beliefs, and/or religious faith predict intentions to vaccinate for COVID-19 in the United States? 

### 1.9. Hypotheses

**H1:** *Positive TP constructs (i.e., Past Positive, Present Hedonistic, and Future), CFC-F, and/or the Balanced Time Perspective (BTP) profile will predict enhanced COVID-19 vaccination intention*.

**H2:** *Negative TP constructs (i.e., Past Negative and Present Fatalistic), CFC-I, and/or the Negative Time Perspective (NTP) profile will predict decreased COVID-19 vaccination intention*.

**H3:** *High scores on measurements of conspiracy beliefs about COVID-19 being a hoax and/or a man-made bioweapon and religious faith will predict decreased COVID-19 vaccination intention*.

## 2. Materials and Methods

### 2.1. Design

The present study followed a cross-sectional design where several questionnaires were administered as outcome measures only once. 

### 2.2. Criteria for Subject Selection

#### 2.2.1. Total Sample

Two hundred thirty-six adult participants living in the United States participated in this study (*n* = 236). Nonetheless, four participants were excluded from the statistical analyses of this study due to excessive missing data. As such, the final sample included two hundred thirty-two adult participants living in the United States (*n* = 232).

#### 2.2.2. Gender of Subjects

Ninety-nine men (*n* = 99), one hundred twenty-nine women (*n* = 129), and two individuals who identified as other gender participated in this study. To address alternative gender identities, the sociodemographic questionnaire part of study included the option of “other gender”. 

#### 2.2.3. Age of Subjects

Adult participants from age 18 upwards participated in this study (M*_age_* = 31).

#### 2.2.4. Inclusion Criteria

Individuals who were 18 years old or older, and either lived in the United States or were born in the United States, were recruited to participate in this study.

#### 2.2.5. Exclusion Criteria

Participants who were under 18 years old and were not born in the United States, as well as did not live in the United States, were excluded from this study. 

### 2.3. Study Advertising, Subject Identification, and Recruitment Strategy

Online advertising was utilized to recruit potential participants from the general population through synchronous methods of recruitment on platforms such as Google Forms and Prolific. Participants recruited through social media platforms such as Facebook were also invited to distribute the link to the study to their contacts (i.e., snowball sampling method). As such, online recruitment for this study was conducted through the snowball method as a convenience sample, since this approach was expected to enhance participant accessibility, as well as racial/ethnic diversity. 

Advertisement text was tailored based on study eligibility criteria. Moreover, advertisements appeared as banners, text, or URL links for users to click on if they were interested in the study. By clicking on an advertisement, the user was directed to our online screening questionnaire. Furthermore, asynchronous methods of recruitment were utilized, involving push technologies such as email and text messages, by using the same texts and images utilized for advertising. 

### 2.4. Study Timeline

Participants were recruited to fill out the corresponding questionnaires online during the fall of the year 2020, as well as the winter and spring of the year 2021. 

### 2.5. Ethical Statement 

The present study was not considered to presents any risks to its participants, since it only involved the online administration of psychosocial questionnaires. Moreover, any potential risks were considered to have been held to a minimum due to the study having been designed to comply with Privacy Rules from the Health Insurance Portability and Accountability Act (HIPPA), the American Psychological Association (APA)’s Ethical Principles of Psychologists and Code of Conduct [64], the 1964 Helsinki Declaration, and its later amendments. Specifically, participants’ confidentiality was safeguarded by keeping all participants’ data de-identified, as well as through anonymous coding. As such, no undue stress or embarrassment was anticipated as a result of online participation. 

### 2.6. Protection against Any Potential Unforeseen Risks

Regarding methods implemented to protect participants against any potential unforeseen risks, on the recruitment materials, participants were provided with the principal investigator’s contact information so that they could address any potential concerns about the study. Moreover, participants were also notified on the recruitment materials that they could withdraw from the study at any time by not completing the surveys with no adverse consequences. As such, the participants of this study were not penalized for choosing at any time point to withdraw from participating in the study. 

### 2.7. Informed Consent

Western Institutional Review Board’s (WIRB’s) Affairs Department reviewed the present study under the Common Rule and applicable guidance (WIRB Work Order Number: 1-1346716-1). WIRB deemed the present study to be exempt under 45 CFR § 46.104(d)(2) because the research only included interactions involving educational tests, survey procedures, interview procedures, or observations of public behavior; and the information obtained is recorded by the investigator in such a manner that the identity of the human subjects cannot readily be ascertained, directly or through identifiers linked to the subjects. Thus, for the present research study, consent was not obtained from participants, since data were maintained in confidentiality and no identifying information was requested from participants on the survey study. In other words, this study involved a consent waiver, also known as a waiver of informed consent. 

### 2.8. Payment for Participation

No compensation was given to participants of the present study. Thus, the recruitment text indicated that participation in this study was completely voluntary.

### 2.9. Race, Ethnicity, and Vaccination for COVID-19 in the United States

One of the most consistent predictors of intention to vaccinate against COVID-19 in the United States is race and ethnicity [65]. For instance, Black Americans have consistently exhibited greater hesitancy to receive COVID-19 vaccines [66,67,68,69]. Moreover, a national probability-based panel survey of adults in the United States called the “*Understanding America Study*” identified Black Americans as having had the lowest probability of receiving a COVID-19 vaccination at the onset of the COVID-19 pandemic, while Asian/Pacific Islanders exhibited the highest probability of getting a vaccination, followed by Whites and Hispanic/Latinos [65]. Nonetheless, the referenced panel survey also revealed that the gap in COVID-19 vaccine intentions between Black Americans and other racial groups widened over time in most cases [65]. The described findings from the *Understanding America Study* bear resemblance to a recent report from the American Association of Retired Persons (AARP), which stated that as of June 2021, the Centers for Disease Control (CDC) and Prevention data identified COVID-19 vaccination rates among race/ethnicity as follows: 60% Whites, 15% Hispanic/Latinos, 9% Black, and 6% Asians in the United States [70]. As such, the role of race and ethnicity on intentions to vaccinate for COVID-19 was addressed in the present study through sociodemographic questions.

### 2.10. Demographic Characteristics of COVID-19 Vaccination Hesitancy in the United States

Beyond race and ethnicity, several other demographic factors have been shown to exert influence on intention to vaccinate for COVID-19 in the United States. For instance, Black and Hispanic Americans, younger adults, females, individuals with lower education qualifications, individuals with lower incomes, and individuals living in rural areas have been identified as being less inclined to receive the COVID-19 vaccine than their counterparts [71,72]. The described variation across demographic groups helps explain why about a third of eligible Americans remain unvaccinated [73]. Thus, demographic factors such as age, gender, education level, income, source of income (i.e., employment/unemployment status), and urbanicity (i.e., urban vs. rural) were addressed in the current study through sociodemographic questions. 

### 2.11. Measurement Outcomes

#### 2.11.1. Sociodemographic questions

Sociodemographic information was collected in this study from participants. Specifically, for this study, participants were asked to disclose information about their race, ethnicity, age, gender, education level, income, source of income (i.e., employment/unemployment status), region of residence (i.e., urban vs. rural), and marital status. Additionally, participants were asked the question: “*Have you been ill with the novel coronavirus/COVID-19?*”, in order to address whether or not the participants of the present study had previously contracted COVID-19. 

#### 2.11.2. Measurement of Intention to Vaccinate for COVID-19, Based on the Theory of Planned Behavior 

For the present study, we measured intention to vaccinate for COVID-19 through the question: “*How likely is it that you are going to have the COVID-19 vaccination*?”. This question was utilized to measure vaccination intention in our study because a similar question was previously utilized to measure vaccination intention for Human Swine influenza (HSI; also known as H1N1) in a previous study based on the Theory of Planned Behavior (Liao et al., 2011). Specifically, in the referenced article vaccination intention was measured through the question: “*How likely is it that you are going to have the HSI vaccination this winter*?”. As such, the only change applied in our study to the referenced question was that the term “*HSI vaccination this winter*?” was replaced with the term “*COVID-19*”. Thus, vaccination intention for COVID-19 was measured in our study through a variation of an equivalent question utilized to measure vaccination intention for HSI/H1N1 in the referenced HSI/H1N1 study [60].

#### 2.11.3. Zimbardo Time Perspective Inventory (ZTPI)

To measure TP across the participants of this study, we administered the 15-item abbreviated version of the ZTPI known as Short ZTPI (i.e., SZTPI), since this version was validated in the United States [74]. The rationale for specifically administering the SZTPI in this study was to reduce the item-burden of participants. In terms of scale design, each item of the ZTPI follows a 5-point Likert scale format, ranging from very characteristic (5) to very uncharacteristic (1). Moreover, the psychometric properties reported for the original validation of the SZTPI in the United States demonstrated adequate internal consistency for all subscales, with α = 0.73 for Past Positive, α = 0.81 for Past Negative, α = 0.78 for Present Hedonistic, α = 0.75 for Present Fatalistic, and α = 0.67 for Future. In our study, assessment of internal consistency yielded similar results, with α = 0.75 for Past Positive, α = 0.86 for Past Negative, α = 0.63 for Present Hedonistic, α = 0.52 for Present Fatalistic, and α = 0.64 for Future. 

Furthermore, we tested the psychometric properties of the applied ZTPI—short version. We conducted Confirmatory Factor Analysis (CFA) to confirm the hypothesized factor–structure on the applied ZTPI—short version [74] (cf. Figure 1). We observed that all item loading of Past Positive, Past Negative, and Future were >0.56. Further, for Present Hedonistic we observed (PH1 = 0.80 and PH2 = 0.57). Similarly, for Present Fatalistic we also observed two out of three items with sufficient high loading (PF2 = 0.78 and PF3 = 0.48). The remaining 2 critical items had relatively low factor loadings (PH3 = 0.33 and PF1 = 0.39). Most important the observed model–fit indices were relatively high (IFI = 0.882; TLI = 816; CFI = 0.877). 

The reliabilities (by Cronbach’s Alpha) were as follows: α = 0.749 (PP); α = 0.856 (PN); α = 0.522 (PF); α = 0.633 (PH) and α = 0.643 (F). For low Cronbach’s Alpha, as in the current analysis for PF and PH, several authors [75,76] argue that a smaller Cronbach’s Alpha value is acceptable in the case of short versions when the number of items is quite small.

In our study, Past Negative was comprised of the following items: *“I think about the bad things that have happened to me in the past”* (PN1); *“Painful past experiences keep being replayed in my mind”* (PN2); and *“It’s hard for me to forget unpleasant images of my youth”* (PN3). Past Positive was comprised of the following items: *“Familiar childhood sights, sounds, smells often bring back a flood of wonderful memories”* (PP1); *“Happy memories of good times spring readily to mind”* (PP2); and *“I enjoy stories about how things used to be in the “good old times”* (PP3). Present Fatalistic was comprised of the following items: *“Life today is too complicated; I would prefer the simpler life of the past”* (PF1); *“Since whatever will be will be, it doesn’t really matter what I do”* (PF2); and *“Often luck pays off better than hard work”* (PF3). Present Hedonistic was comprised of the following items: *“I make decisions on the spur of the moment”* (PH1); *“Taking risks keeps my life from becoming boring” (PH2); and “It is important to put excitement in my life”* (PH3). Future was comprised of the following items: *“When I want to achieve something, I set goals and consider specific means for reaching those goals”* (F1); *“Meeting tomorrow’s deadlines and doing other necessary work comes before tonight’s play”* (F2); *and “I complete projects on time by making steady progress”* (F3). 

To measure BTP and NTP profiles, deviation coefficients were calculated in this study. First, the Deviation from the Balanced Time Perspective revisited (DBTPr) coefficient was calculated to measure the NTP profile [77]. This coefficient measures the extent to which individuals deviate from an ideal TP profile, where the farther a DBTP value is from zero, the more misbalanced the individual’s TP profile would be considered to be, resulting in the NTP profile.

In this calculation, e equals the expected optimal value for each TP, as stipulated by Jankowski, Zajenkowski, and Stolarski [1.0 for Past Negative (PN), 5.0 for Past Positive (PP), 1.0 for Present Fatalistic (PF), 3.4 for Present Hedonistic (PH), and 5.0 for Future (F)]. 

Contrary to the DBTPr, the Deviation from the Negative Time Perspective (DNTP) coefficient measures the extent to which individuals deviate from a misbalanced TP profile [29]. As such, the farther a value would appear to be from zero, the more optimal the individual’s TP profile would be considered to be. 

In this calculation, n equals the observed negative value obtained for each measured TP, whereas e equals the expected negative value for each TP, as stipulated by Zimbardo, Sword and Sword [4.35 for Past Negative (PN), 2.80 for Past Positive (PP), 3.30 for Present Fatalistic (PF), 2.65 for Present Hedonistic (PH), and 2.75 for Future (F)] [31].

#### 2.11.4. Consideration of Future Consequences (CFC) Scale

To measure Consideration of Future Consequences (CFC) in our study, we administered an ultra-short version of the scale comprising 6 items [78]. In terms of scale design, each item of the CFC follows a 7-point Likert scale format, ranging from “extremely characteristic of you” (7) to very “extremely uncharacteristic of you” (1). Despite the original view of CFC being unidimensional, following the presumption that immediate consequences and considerations for future represent opposite ends within the same dimension [37], for this study, we followed the theoretical framework that views this scale as two-dimensional, since within the past decade, scholars have re-examined the CFC and concluded that it is rather two-dimensional [79,80]. According to this two-dimensional view, CFC comprises an immediate factor (labeled CFC-I), as well as a future factor (labeled CFC-F). Thus, the described two-factor model proposes the idea that concerns about short- and long-term outcomes are distinct rather than opposites. As such, in our study we accounted for an immediate factor (labeled CFC-I), as well as a future factor (labeled CFC-F). Regarding psychometric properties, in our study the CFC scale demonstrated adequate internal consistency for all subscales, with α = 0.84 for CFC-I and α = 0.73 for CFC-F. The CFC-I factor in our study was comprised of the following items: “*I only act to satisfy immediate concerns, figuring the future will take care of itself*”; “*I think that sacrificing now is usually unnecessary since future outcomes can be dealt with at a later time*”; and “*I only act to satisfy immediate concerns, figuring that I will take care of future problems that may occur at a later date*”. On the other hand, the CFC-F factor in our study was comprised of the following items: “*I consider how things might be in the future, and try to influence those things with my day to day behavior*”; “*When I make a decision, I think about how it might affect me in the future*”; and “*My behavior is generally influenced by future consequences*”.

#### 2.11.5. The COVID-19 Conspiracy Beliefs Questionnaire

To measure conspiracy beliefs about the COVID-19 pandemic, we administered a 6-item COVID-19 Conspiracy Beliefs questionnaire published during the first year of the COVID-19 pandemic [81]. All six items follow a Likert scale format, ranging from 1 (strongly disagree) to 7 (strongly agree). This questionnaire was selected for the present study due to it tapping into the two most prevalent conspiracy beliefs: a) the idea that COVID-19 is a harmless virus that receives overblown attention for the personal benefit of a few people (i.e., COVID-19 is a hoax); and b) the logically incompatible idea that COVID-19 was purposefully created for the personal benefit of some individuals (i.e., COVID-19 as a human-made bioweapon). As such, the described COVID-19 Conspiracy Beliefs questionnaire entails a two-factor model, where each factor is composed of 3 items. In our study, the described COVID-19 Conspiracy Beliefs questionnaire demonstrated adequate internal consistency for all subscales, with α = 0.56 for belief about COVID-19 being a hoax and α = 0.69 for belief about COVID-19 being a human-made bioweapon.

For the next step, we conducted a CFA for the two-factorial model of the COVID-19 Conspiracy Beliefs questionnaire (cf. Figure 2) [81]. Two items (one of each of the assumed factors “hoax” and “bioweapon”) showed a low factor-loading (0.19 and 0.27). For all other items, we observed excellent loadings of >0.88. The model fit indices reached high values despite the two critical items, indicating a good model fit (IFI = 0.972; TLI = 0.927; CFI = 0.972). The inter-correlation reached a relatively high positive value (r = 0.66), which is in line with the theoretical consideration by Imhoff and Lamberty [81], reflecting the assumed “illogical” response tendency of participants scoring high on both factors at the same time.

The *“COVID-19 is a hoax”* factor in our study was comprised of the following items: *“The virus is intentionally presented as dangerous in order to mislead the public”; “Experts intentionally mislead us for their own benefit, even though the virus is not worse than a flu”;* and *“We should believe experts when they say that the virus is dangerous”*. On the other hand, the *“COVID-19 as a human-made bioweapon”* factor in our study was comprised of the following items: *“Corona was intentionally brought into the world to reduce the population”; “Dark forces want to use the virus to rule the world”;* and *“I think it’s nonsense that the virus was created in a laboratory”.*

#### 2.11.6. The Santa Clara Strength of Religious Faith Questionnaire

In order to measure religious faith in our study, we administered the final item of the brief, 5-item version of the Santa Clara Strength of Religious Faith Questionnaire (SCSRFQ) [82]. The described final SCSRFQ item reads as follows: “*My faith impacts many of my decisions*”. This item, like all five items of the SCSRFQ, follows a Likert scale format, ranging from 1 (strongly disagree) to 4 (strongly agree). Specifically, the referenced item was chosen for the present study because it addresses the extent to which religious faith impacts decision-making. Hence, since the intention to vaccinate represents a decision-making process, the described item was chosen out of all SCSRFQ items due to it being the one bearing most relevance for our study, which is focused on addressing factors influencing vaccination decisions.

### 2.12. Statistical Analyses

Regarding descriptive statistics, for this study correlation analyses were conducted for the main predictor independent variables (IV), which include: all five TP sub-scales (Past Positive, Past Negative, Present Hedonistic, Present Fatalistic, and Future), the DBTPr and DNTP coefficients, CFC-I, CFC-F, the “*COVID-19 is a hoax*” variable, the “*COVID-19 is a human-made bioweapon*” variable, the SCSRFQ item: “*My faith impacts many of my decisions*”, and the dependent variable (DV): COVID-19 vaccination intention based on TPB (see Table 1). 

To test our hypotheses, hierarchical linear regression analyses (HLRAs) were conducted to examine the influence of the controlling variables and main predictor variables of TPs, DBTPr, DNTP, CFC-I, CFC-F, “*COVID-19 is a hoax*”, “*COVID-19 is a human-made bioweapon*”, and each individual SCSRFQ item on the outcome variable of vaccination intention, based on TPB (see Table 2). The first block included age, gender, education level, income, source of income (i.e., employment/unemployment status), urbanicity (i.e., urban vs. rural), marital status, and previous history of contagion with COVID-19 as controlling variables, utilizing the enter method. The second block included race (as a set of dummy variables with dummy 1 = *Black/African American*; dummy 2 = *Asian/Pacific*; dummy 3 = *Caucasian,* and dummy 4 = *Multiracial*/*Mixed)/Other*); as well as one category for ethnicity: (i.e., *Hispanic/Latino*) as controlling variables, utilizing the stepwise method. The third block included the five TP tendencies (Past Positive, Past Negative, Present Hedonistic, Present Fatalistic, and Future) as predictors, utilizing the enter method. The fourth block included the DBTPr and DNTP coefficients as predictors, utilizing the enter method. The fifth block included CFC-I and CFC-F as predictors, utilizing the enter method. The sixth block included the variables “*COVID-19 is a hoax*”, and “*COVID-19 is a human-made bioweapon*” as predictors, utilizing the enter method. The seventh block included the SCSRFQ item: “*My faith impacts many of my decisions*” as a predictor, utilizing the enter method. The described seventh-block approach was implemented to measure the additive predictive value of each variable assessed through the third, fourth, fifth, sixth, and seventh blocks, beyond the predictive value of the controlling variables assessed through the first and second blocks. Following the described methods, one HLRA was conducted on the outcome variable of vaccination intention based on TPB to test our hypotheses. Based on a conservative assumption of a modest small effect size of Cohen’s f^2^ = 0.05 [83], we conducted a G-Power analysis [84] with 12 predictors, resulting in a sample size of *n* = 218.

## 3. Results

To test our hypotheses on the intention to vaccinate against COVID-19, we applied a Hierarchical Linear Regression Analysis (HLRA) including all five TP dimensions (Past Positive, Past Negative, Present Hedonistic, Present Fatalistic, and Future); Deviation from a Balanced Time Perspective—revisited (DBTP-r); Deviation from a Negative Time Perspective (DNTP); the two factors of the Consider Future Consequences Scale (CFC): immediate (CFC-I), and future (CFC-F); belief in COVID-19 as hoax; belief in COVID-19 as bioweapon, and one item of the SCSRFQ as predictors. The following variables were included as control variables: age, gender, marital status, urbanicity (i.e., urban vs. rural), race (as a set of dummy variables with dummy 1 = *Black/African American*; dummy 2 = *Asian/Pacific*; dummy 3 = *Caucasian,* and dummy 4 = *Multiracial*/*Mixed/Other*), ethnicity (i.e., *Hispanic/Latino*), highest level of education, and previous history of contagion with COVID-19. We applied the following blocks: Block 1 included age, gender, marital status, urbanicity, education level, income, employment status, and previous history of contagion with COVID-19 using the enter method. Block 2 included the set of dummy variables of the racial background using the stepwise method. Block 3 included Past Negative, Past Positive, Present Fatalistic, Present Hedonistic, and Future using the enter method. Block 4 included DBTP-r and DNTP using the enter method. Block 5 included CFC-I and CFC-F using the stepwise method. Block 6 included the COVID-19 conspiracy belief factors hoax and bioweapon using the enter method. Finally, Block 7 included the religiosity item “*My faith impacts many of my decisions*.” using the enter method.

The multiple hierarchical regression analyses yielded seven models, with model 7 providing the best model fit (R^2^ change = 0.014; *p* = 0.025) and the highest R and R^2^ values (R = 0.721; R^2^ = 0.519). To ensure transparency, we report in Table 3 the detailed model fit comparisons. The comparisons show that model 7 has the highest model fit (R = 0.689; R^2^ = 0.474). All other models (model 6 to model 1) show a smaller decreasing model fit (all Rs < 0.464; all R^2^ < 0.411). In model 7, only four variables were excluded (dummy 1, 2, 3, and 5). From model 6 to 2, an increasing number of predictors were excluded, resulting in a lower decreasing model fit. Finally, model 1 excluded all stepwise entered predictors, resulting in the lowest model fit (cf. Table 3). 

According to the results (cf. Table 2), we observed the following significant effects on the COVID-19 vaccination intention for model 7: Past Negative had an enhancing effect on the vaccination intention (*β* = 0.486, *B* = 0.856, *SE B* = 0.261, *t* = 3.283, *p* = 0.001), whereas Past Positive reduced the vaccination intention (*β* = −0.252, *B* = −0.547, *SE B* = 0.201, *t* = −2.722, *p =* 0.007). DBTP-r reduced the vaccination intention (*β* = −0.609, *B* = −1.162, *SE B* = 0.366, *t* = −3.174, *p =* 0.002). Furthermore, CFC-F (*β* = 0.356, *B* = 0.689, *SE B* = 0.127, *t* = 5.438, *p <* 0.001) and CFC-I (*β* = 0.169, *B* = 0.237, *SE B* = 0.098, *t* = 2.415, *p =* 0.017) significantly enhanced the vaccination intention. The belief in COVID-19 as a hoax reduced the vaccination intention (*β* = −0.358, *B* = −0.453, *SE B* = 0.083, *t* = −5.455, *p <* 0.001). Finally, approval of the religiosity item “*My faith impacts many of my decisions”* reduced the vaccination intention (*β* = −0.135, *B* = −0.228, *SE B* = 0.108, *t* = −2.104, *p* = 0.037). All other predictors were either non-significant or excluded from the model. Among the control variables, the following reached significance: gender (*β* = −0.147, *B* = −0.572, *SE B* = 0.220, *t* = −2.597, *p* = 0.010), with women showing a reduced vaccination intention; and the dummy 4 variable (*β* = −0.197, *B* = −1.166, *SE B* = 0.315, *t* = −3.696, *p* < 0.001) with participants of mixed or other races showing a reduced vaccination intention. 

## 4. Discussion

In summary, the present study assessed for the first time in the United States the role of all five traditional TP tendencies and TP profiles as predictors of intention to vaccinate against COVID-19 according to TBP, while also accounting for other variables previously shown to also influence intention to vaccinate for COVID-19, such as CFC, conspiracy beliefs, and religiosity. The HLRA conducted in our study on the outcome variable of vaccination intention based on Theory of Planned Behavior following the described methods yielded statistically significant results, both with control variables, as well as with the main variables of our study. In terms of results streaming from the main predictors assessed in our study, the described HLRA revealed that in our study, intention to vaccinate against COVID-19 was mainly predicted by past TP constructs such as Past Positive and Past Negative, CFC-F; CFC-I, the DBTP-r coefficient; belief in COVID-19 as hoax, and by approval of the religiosity item “*My faith impacts many of my decisions*”. 

Our hypotheses about Time Perspectives (TPs) CFC-F, CFC-I, and/or BTP (measured through DBTP-r and DNTP) predicting the intention to vaccinate against COVID-19 was only confirmed partly. Contrary to our hypotheses, Past Negative enhanced vaccination intention, whereas Past Positive reduced the vaccination intention. One preliminary explanation is that individuals scoring high on Past Negative feel more vulnerable to external threats like the COVID-19 virus, and thus show a higher motivation to get vaccinated to protect themselves. In turn, those scoring high on Past Positive might be less threatened by the COVID-19 virus. These assumptions need, however, to be tested further in future research studies. Another important finding is that DBTP-r reduced the vaccination intention, where participants showing deviations from a Balanced Time Perspective appeared to be less motivated to seek vaccination against COVID-19. Accordingly, this finding clearly supported indirectly our hypotheses about the enhancing effects of positive and beneficial TP constructs and reducing effects of negative and harming TP constructs because the BTP is characterized by high beneficial TP constructs and by low harming TP constructs. To re-test this assumption, we repeated the same HLRA without DBTP-R and DNTP. We also removed CFC-F and CFC-I because we assumed a certain overlap between CFC-I and Present Hedonistic and between CFC-F and Future. We observed the following significant results for the Time Perspectives (TPs): As in the first HLRA, we observed the significant enhancing effect of Past Negative (*β* = 0.142, *B* = 0.249, *SE B* = 0.110, *t* = 2.274, *p =* 0.024). Furthermore, Present Hedonistic (*β* = 0.159, *B* = 0.386, *SE B* = 0.151, *t* = 2.560, *p =* 0.011) and Future (*β* = 0.123, *B* = 0.331, *SE B* = 0.165, *t* = 2.010, *p <* 0.001) significantly enhanced the vaccination intention, whereas Present Fatalistic (*β* = −0.143, *B* = −0.329, *SE B* = 0.149, *t* = −2.201, *p =* 0.029) showed a reducing effect on the vaccination intention. In contrast, Past Positive did not reach significance in the second HLRA (*β* = −0.031, *B* = −0.068, *SE B* = 0.138, *t* = −0.491, *p =* 0.624). Taking all these results together we can conclude that our assumption of an indirect influence of the single TPs via DBTP-r (and partly also via CFC-I and CFC-F), which is hidden if these variables are entered into the HLRA, is plausible. All other observations remained the same as in the former HLRA, according to being significant vs. non-significant, or according to the observed directions of influence. 

The findings of both CFC-I and CFC-F having predicted enhanced intention to vaccinate against COVID-19 are supported according to the CFC-F by another study, where CFC-F predicted vaccination intentions against COVID-19 and seasonal flu [42]. Thus, in our study, as well as in a similar study, future-tense TPs appear to be effective predictors for intention to vaccinate against COVID-19. The second finding that CFC-I also enhanced the vaccination intention is contradictory to our hypotheses and needs further explanation. One explanation for this finding could be that individuals characterized by CFC-I would have an enhanced willingness to get vaccinated, in order to ensure they would protect their health as quickly as possible from COVID-19, so that in that way they could immediately continue to proceed with pleasantly enjoying their lives. 

Our third hypothesis regarding conspiracy beliefs about COVID-19 being a hoax and/or a human-made bioweapon, and religious faith predicting decreased intention to vaccinate against COVID-19, was only confirmed with respect to the belief in COVID-19 as a hoax, and the religiosity item addressing how religious faith impacts decisions. The finding of belief in COVID-19 as hoax having predicted reduced intention to vaccinate against COVID-19 could be interpreted as indicating that individuals characterized by this belief clearly are not willing to vaccinate themselves against COVID-19 due to not believing in the seriousness and danger that COVID-19 entails. This finding about belief in COVID-19 as hoax having predicted reduced intention to vaccinate against COVID-19 is supported by systematic reviews and meta-analyses where belief in conspiracy theories about COVID-19 have appeared to be a persistent barrier to COVID-19 vaccine uptake in the United States [43], as well as globally [48,49,50,51]. Furthermore, the finding that a strong influence of faith on one’s decisions reduced the vaccination intention could be interpreted as suggestive that religious faith prompts religious individuals to rely more on their faith than on seeking vaccination for COVID-19, perhaps due to the perception that vaccines are more associated with science than religion. This finding is supported by the results of a study where greater religiosity was associated with being uncertain or unwilling to get the COVID-19 vaccine among adults in Puerto Rico [52]. 

When interpreting the findings observed in our study with Past Negative and Past Positive, it is important to consider that this is the first time that a research study has reported associations between past-tensed TPs and intention to vaccinate against COVID-19. For instance, the few existing studies on TP-like constructs and vaccination intentions against COVID-19 focused on specific present-tensed and/or future-tensed TPs [15,41,42]. Thus, the present study further extends knowledge about TP’s relationship with vaccination intentions against COVID-19, since it uncovers for the first time ever that past-tensed TPs predict vaccination intentions against COVID-19.

Regarding the lack of statistically significant findings with Present Hedonistic, Present Fatalistic, and Future in our study, it can be inferred that their influence is indirect via the DBTP-r. This inference is supported by the statistically significant findings observed through the second additional HLRA conducted for the present study. 

When interpreting the results of our study, it is important to note that only two of the controlling variables accounted for in our study yielded statistically significant findings through the HLRA conducted in our study: gender and self-identification as multiracial/mixed. Regarding gender, female participants showed a lower vaccination intention. Concerning race, individuals identified as multiracial or from mixed origin exhibited a reduced vaccination intention. All other control variables yielded non-significant results. 

### 4.1. Limitations and Future Directions

Regarding limitations, we can highlight that the present study was cross-sectional, and relied on self-reported data rather than other measurement approaches, such as behavioral tasks. Thus, we acknowledge that it is possible that fear of self-disclosure and stigma could have affected the results of this study. Moreover, in our study, TP, CFC, and religious faith were measured through abbreviated versions of the questionnaires developed to assess those constructs. As such, our results should be interpreted cautiously. 

Furthermore, the sample was not planned as a representative sample of the United States’ population. Since the current population of the United States is estimated to be about 334.3 million individuals, and the final sample was *n* = 232, it is important to interpret our results cautiously against the background of such a large country, characterized by such a diverse population. As such, this is a clear limitation of the current study. We do, however, not assume explicitly substantial differences of the association between time perspectives and attitudes or actual behavior towards COVID-19 vaccination. What can be assumed is that time perspectives differ across cultures [85,86], sociodemographic variables [25,87], age [88,89], race [90], ethnicity [91], education [92,93,94,95], income [96], and urbanicity [91] of the overall population. Against this background, we encourage future studies with representative samples for re-testing our results in a broader framing.

In terms of future directions, we acknowledge that several factors exist that could potentially influence vaccination intention against COVID-19 beyond TP, BTP, CFC, conspiracy beliefs about COVID-19, and religious faith. Some of these other factors include health-related behaviors and specific beliefs concerning COVID-19 and its vaccines, as well as general beliefs, such as fear of contamination, intolerance of uncertainty, emotional states, coping behavior, and the source of information concerning the virus [97]. Nonetheless, these factors were not measured in this study, which by design was focused on the constructs of vaccination intention against COVID-19, TP, CFC, conspiracy beliefs about COVID-19, and religious faith. Thus, the potential influence of any of these factors on vaccination intention against COVID-19, TP, BTP, CFC, conspiracy beliefs about COVID-19, and/or religious faith as mediators or moderators reflects interesting avenues of future research that extend beyond the scope of the present study.

### 4.2. Implication of the Study

Although this study did not directly benefit its participants, we hope that the results of this study will assist with the education about, prevention of, and intervention against COVID-19. Benefits of this study include providing the public health field a better understanding of the impact of utilizing education, prevention, and intervention strategies against COVID-19. This information would be beneficial to transfer the knowledge acquired through this study to behavioral interventions, health promotion campaigns, and the academic field.

## 5. Conclusions

Our study extended previous research by showing that the intention to vaccinate against COVID-19 based on the Theory of Planned Behavior was mainly predicted by past-TP constructs such as Past Positive and Past Negative, the BTP profile measured through the DBTP-r coefficient, CFC-F and CFC-I, belief in COVID-19 as hoax, and religious faith as having a strong influence on one’s vaccination decisions. Specifically, vaccination intention was reduced by gender identification as woman, identification as multiracial or from mixed origin, Past Positive, Deviation from BTP (i.e., a high DBTP-r), belief in COVID-19 as hoax, and religious faith. Conversely, intention to vaccinate against COVID-19 was increased by Past Negative, CFC-F, and CFC-I. In particular, one of the most striking findings of our study is that a Balanced Time Perspective appears to exert a positive influence on enhancing COVID-19 vaccination intention, as deducted by how Deviation from BTP reduces intention to vaccinate against COVID-19. These findings could be beneficial for knowledge transfer to behavioral interventions aimed at promoting vaccination against COVID-19, health promotion campaigns, and the public health field.

## Figures and Tables

**Figure 1 ijerph-20-03625-f001:**
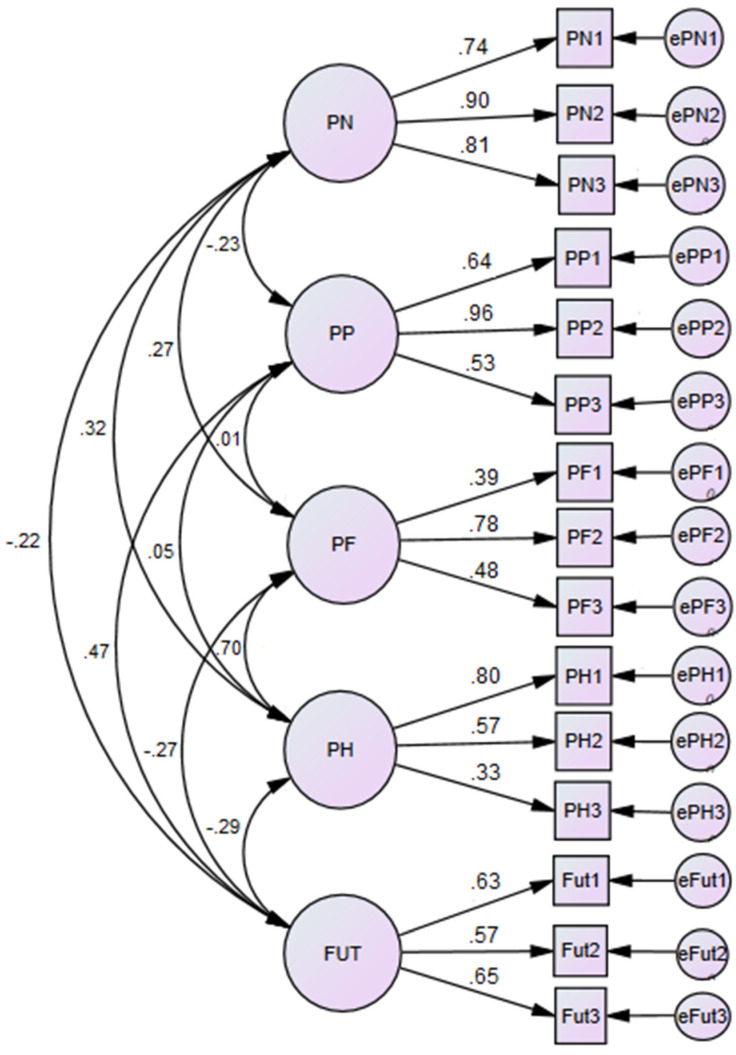
Confirmatory Factor Analysis (CFA) for the Short version of the Zimbardo Time Perspective Inventory (ZTPI).

**Figure 2 ijerph-20-03625-f002:**
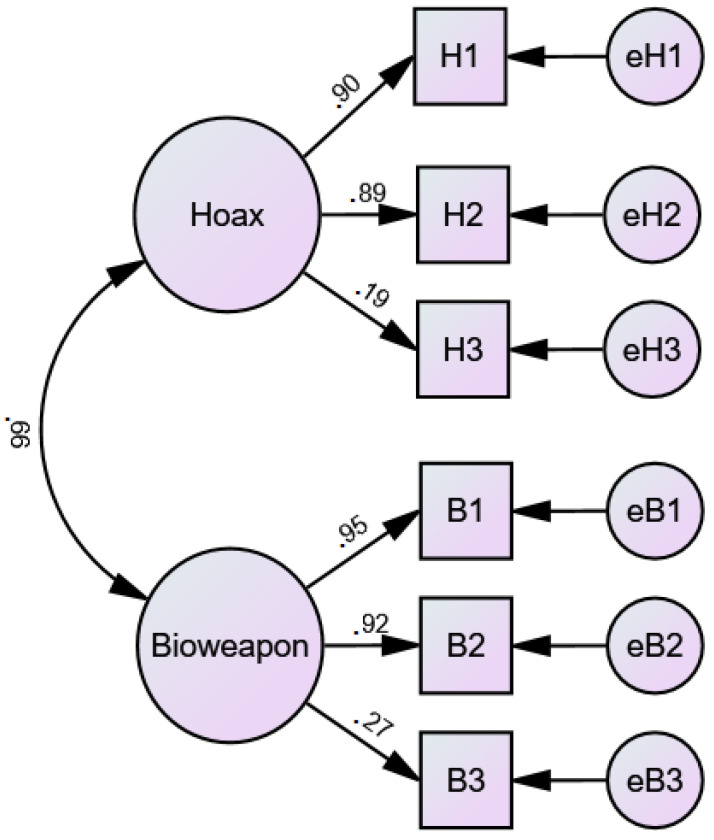
Confirmatory Factor Analysis (CFA) for the two-factorial COVID-19 Conspiracy Beliefs Scale.

**Table 1 ijerph-20-03625-t001:** Pearson’s correlations among TP subscales, DBTP-r, DNTP, CFC factors, COVID-19 as hoax, COVID-19 as bioweapon, race (as 4 dummy variables), one item measuring religiosity, and vaccination intention.

	1	2	3	4	5	6	7	8	9	10	11	12	13	14	15	16	17
Past Negative	-																
Past Positive	−0.149 *	-															
Present Fatalistic	0.267 **	0.071	-														
Present Hedonistic	0.228 **	0.099	0.331 **	-													
Future	−0.144 *	0.285 **	−0.145 *	−0.102	-												
DBTP-R	0.795 **	−0.460 **	0.487 **	0.156 *	−0.451 **	-											
DNTP	−0.473 **	0.330 **	−0.324 **	0.036	0.402 **	0.532 **	-										
CFC Future	−0.080	0.173 **	−0.347 **	−0.216 **	0.378 **	−0.249 **	0.222 **	-									
CFC Immediate	0.162 *	0.008	0.496 **	0.464 **	−0.187 **	0.274 **	−0.140 **	−0.422 **	-								
COVID is a Hoax	−0.146 *	0.210 **	0.192 **	.161 *	0.155 *	−0.130 *	0.166 *	−0.119	0.243 **	-							
COVID = Bioweapon	−0.041	0.264 **	0.269 **	0.150 *	0.064	−0.003	0.093	−0.137 *	0.297 **	0.499 **	-						
Dummy 1	0.075	0.071	0.001	0.026	−0.041	0.034	−0.032	0.033	−0.096	−0.056	0.027	-					
Dummy 2	0.049	0.037	0.138 *	0.097	−0.057	0.056	−0.004	−0.064	0.125	−0.035	0.093	−0.142 *	-				
Dummy 3	0.087	−0.203 **	−0.006	−0.098	−0.051	0.111	−0.143 *	−0.081	−0.003	−0.144 *	0.147 *	−0.294 **	−0.469 **	-			
Dummy 4	0.094	0.167 *	−0.077	−0.083	0.020	−0.129	0.076	0.063	−0.071	0.039	0.008	−0.113 *	−0.180 **	−0.372 **	-		
Religiosity Item	−0.140	0.276 **	−0.059	0.032	0.112	−0.231 **	0.103	0.189 **	0.049	0.357 **	0.360 **	0.151 *	−0.045	−0.129	−0.025	-	
Vaccination Intention	−0.210 **	−0.182 **	−0.132 *	0.094	0.013	0.091	−0.133 *	0.265 **	−0.048	−0.459 **	−0.325 **	−0.018	0.220 **	0.044	−0.220 **	−0.226 **	-
*Mean*	3.155	3.644	2.523	3.052	3.892	3.628	1.650	5.554	2.975	2.782	2.588	0.082	0.185	0.491	0.125	2.250	5.017
*SD*	1.129	0.913	0.871	0.818	0.742	1.048	0.383	1.021	1.413	1.562	1.487	0.275	0.389	0.501	0.331	1.176	1.987

Notes: TP = Time Perspective; DBTP-r = Deviation from Balanced Time Perspective—revisited; DNTP = Deviation from Negative Time Perspective; CFC = Consideration of Future Consequences; Dummy 1 = Black/African American; Dummy 2 = Asian/Pacific/Alaska/Native American; Dummy 3 = Caucasian/European; Dummy 4 = Mixed/Others; Religiosity Item = “*My faith impacts many of my decisions*.” * *p* < 0.05, ** *p* < 0.01.

**Table 2 ijerph-20-03625-t002:** Hierarchical linear regression run with control variables, Time Perspectives (TPs), DBTP-r, DNTP, Consideration of Future Consequences (CFC) factors, conspiracy beliefs in COVID-19 as a hoax or bioweapon, and religious faith as predictors of COVID-19 vaccination intention.

	*Β*	*β*	Sig. *β*	f^2^	*R*	*R^2^*	*Δ* *R* * ^2^ *	*Δ* *F*	Sig. *ΔF*
Model 7					0.689	0.474	0.012	4.426	0.037
Age	−0.056	−0.040	0.516	0.002				−0.040	−0.040
Gender	−0.572	−0.147	0.010	0.025				−0.147	−0.147
Marital Status	0.465	0.116	0.069	0.015				0.116	0.116
Urbanicity/Rurality	0.149	0.030	0.581	0.000				0.030	0.030
Highest Education Level	0.121	0.117	0.062	0.044				0.117	0.117
Annual Income	−0.058	−0.112	0.131	0.036				−0.112	−0.112
Employment Status	−0.019	−0.020	0.748	0.000				−0.020	−0.020
Prior COVID−19 Contagion	0.304	0.080	0.128	0.011				0.080	0.080
Dummy 4 (Mixed)	−1.166	−0.197	<0.001	0.067				−0.197	−0.197
Past Negative	0.856	0.486	0.001	0.053				0.486	0.486
Past Positive	−0.547	−0.252	0.007	0.055				−0.252	−0.252
Present Fatalistic	0.160	0.070	0.476	0.002				0.070	0.070
Present Hedonistic	0.226	0.092	0.172	0.008				0.092	0.092
Future	−0.243	−0.090	0.279	0.006				−0.090	−0.090
DBTP-r	−1.162	−0.609	0.002	0.049				−0.609	−0.609
DNTP	−0.236	−0.046	0.538	0.002				−0.046	−0.046
CFC-I	0.237	0.169	0.017	0.146				0.169	0.169
CFC-F	0.689	0.356	<0.001	0.029				0.356	0.356
COVID-19 Hoax	−0.453	−0.358	<0.001	0.146				−0.358	−0.358
COVID-19 Bioweapon	−0.045	−0.034	0.612	0.000				−0.034	−0.034
Religious Faith Item 5	−0.228	−0.135	0.037	0.021				−0.135	−0.135

Notes: Dummy 4 = Mixed/Others; DBTP-R = Deviation Balanced Time Perspective coefficient revisited; DNTP = Deviation Negative Time Perspective; CFC-I = Consideration of Future Consequences—Immediate; CFC-F = Consideration of Future Consequences—Future; Religious Faith Item # 5 from the Santa Clara Strength of Religious Faith Questionnaire (SCSRFQ) = “*My faith impacts many of my decisions*”. * According to Cohen (1992), the reported effect sizes (i.e., f^2^) can be interpreted as: 0.02 = small; 0.15 = middle; 0.35 = strong.

**Table 3 ijerph-20-03625-t003:** Model Summary of the HLRA.

					Change Statistics				
Model	R	R^2^	Adjusted R^2^	SE of the Estimate	R^2^ Change	F Change	df1	df2	Sig. F Change
1	0.260	0.068	0.033	1.9579	0.068	1.944	8	214	0.055
2	0.344	0.119	0.081	1.9082	0.051	12.278	1	213	0.001
3	0.458	0.210	0.156	1.8285	0.091	4.795	5	208	0.000
4	0.487	0.237	0.177	1.8057	0.027	3.648	2	206	0.028
5	0.575	0.331	0.271	1.6993	0.094	14.292	2	204	0.000
6	0.680	0.463	0.410	1.5296	0.132	24.900	2	202	0.000
7	0.689	0.474	0.420	1.5168	0.012	4.426	1	201	0.037

## Data Availability

Source data for tables and figures are provided with the paper.

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
