# Peer review of "A Time to Get Vaccinated? The Role of Time Perspective, Consideration of Future Consequences, Conspiracy Beliefs, Religious Faith, Gender, and Race on Intention to Vaccinate for COVID-19 in the United States"

_ijerph, 2023, doi:10.3390/ijerph20043625_

Round 1

Reviewer 1 Report

Understanding how individuals decide to vaccinate against Covid-19 is a topic of the utmost importance. However, the manuscript lacks an overarching theoretical perspective. As written, it is like a “competition” between theory of planned behavior and time perspective. Clarifying which theory is actually framing this work would be beneficial to the manuscript. This would also help with the analytic strategy. Moreover, several subscales accessing constructs that are key to the manuscript are not reliable. Lastly, the literature review could be more focused, including consideration of the examination of race and religion. Further details (in rank of importance) are provided below.

1.       While the ideas espoused by the ZPTI have truly changed the field, there have been more than a dozen studies critiquing the ZTPI’s psychometric qualities, including its structure and reliability. Thus, the literature is not accurately described. Relatedly, subscales of the ZTPI and Covid Beliefs measures yielded scores that were not reliable, as internal consistency estimates were below the field’s convention of .70. This should be addressed.

3.      Regarding the analytic strategy, it weakens the manuscript that 7 models were run, but only 1 model was reported. In the interest of transparency, it would be helpful to understand the iterations that were observed across models. Further, if time perspective is the primary focus of the manuscript, it was unclear why it would be included in a block with the CFC. Developing the theoretical framing would also be helpful for guiding decisions about the analyses.

4.       Examining race-related differences in vaccination without considering the strong confound between race and income is an inaccurate portrayal of the topic. Race has nuanced relationships with health behaviors and access to health care. Given the primary focus of the paper, it does not make sense to include it as a topic of investigation. If the authors are interested in the topic of race, a more thorough consideration and review of the literature would be warranted.

5.    The demographic variables (race, religion, income, etc..) might be better suited as controls. In this way, they would be moved to the method section and not included in the literature review.

6.       Literature review section on 1.3 could be expanded. What did the studies reported in this section find regarding TBP and vaccines for other illnesses?

7.       The literature on time perspective included research that was superfluous to the aims of the current study. This section should be revised extensively. For example, it is not clear why a study about food addiction would support a study about Covid vaccination.

8.       Were the procedures approved by an Institutional Review Board? This should be stated and a study number included.

9.       For all the measures, response options should be included.

10.   It is unclear why the equation is reported when published elsewhere. This could be cited for the reader’s reference.

11.   Effect size should be reported and interpreted.

Reviewer 2 Report

The present study examined the predictability of Time Perspective (TP) tendencies (i.e: Past Positive, Past Negative, Present Hedonistic, Present Fatalistic, and Future), the Balanced Time Perspective (BTP) profile, the Consideration of Future Consequences-Immediate (CFC-I) factor, the Consideration of Future Consequences-Future (CFC-F) factor, conspiracy beliefs about Covid-19 being a hoax, religious faith, gender and race on intention to vaccinate against Covid-19 in the United States, according to the Theory of Planned Behavior. The idea of the study is good but its application is not valid. The construction of the questionnaire is not reported and the final sample used in the current study was small and not representative for the population. Further, there was differences in the total sample size based on gender (N=230) compared to what reported in the abstract and methodology (N=232). The sample size calculation should be provided and clarify how can this size sufficient to conduct this study since an online questionnaire was used by Prolific and Google Forms. Very view results are obtained from this study which are not significant to the published literature. Introduction looks like a review article and it is very long. It should be summarized without subdivisions.

Round 2

Reviewer 1 Report

Thank you for the opportunity to review the revised version of the manuscript entitled “A Time to Get Vaccinated? The Role of Time Perspective, Consideration of Future Consequences, Conspiracy Beliefs, Religious Faith, Gender and Race on Intention to Vaccinate for Covid-19 in the United States.” The authors made some changes to the original version. However, the primary issue with the conceptualization of the manuscript remains. Many of the major issues raised were not addressed.

Conceptualization

1.       Theoretical framing continues to lack focus.

a.       The manuscript continues to lack a guiding theory. Even though the authors stated that time perspective was now “the overarching theoretical perspective.” The manuscript continues to be framed by Theory of Planned Behavior, Time perspective, and Consideration of Future Consequences. It is unfortunate that the authors did not revise the manuscript’s theoretical framing.

2.       Sociodemographic topics lack supporting evidence.

a.       In this revision, the authors include sociodemographics as a Research Question but have moved the supporting literature to the Method section.

b.       Further, the evidence for examining each of the specific sociodemographic variables is superficial. If the authors keep sociodemographic as a research question, then the literature review should be revised substantially.

Methods/Figures

1.       Tables and Figures were absent.

2.       Effect size (not power estimates) should be reported for the findings.

Abstract

1.       The abstract no longer reflects the conceptualization of the study and should be revised accordingly.  

Other

1.       The writing could be edited. Some sections included only a few sentences (e.g., the first two headers).

2.       The use of acronyms (TPB) was cumbersome. Whole words are preferred for readability.

Reviewer 2 Report

The authors tried to improve the manuscript, however, the major issues are still available including very long introduction with more than 65 references which look like a review article. no need to write 12 pages to show the rational of conducting this study. Furthermore, the sections that were moved to methodology increase nothing to this section. Even with using G-power, the sample size is not adequate for conducting an online questionnaire in a population of 250 millions.

Finally, the intention for covid-19 vaccines has been extensively studies, what are the adding value of this research. what are the variables that make it important. 

Tables are not presented in the revised manuscript. 
